# Potential of Oil Palm Empty Fruit Bunch Resources in Nanocellulose Hydrogel Production for Versatile Applications: A Review

**DOI:** 10.3390/ma13051245

**Published:** 2020-03-10

**Authors:** Farah Nadia Mohammad Padzil, Seng Hua Lee, Zuriyati Mohamed Asa’ari Ainun, Ching Hao Lee, Luqman Chuah Abdullah

**Affiliations:** 1Laboratory of Biopolymer and Derivatives, Institute of Tropical Forestry and Forest Products (INTROP), Universiti Putra Malaysia, Serdang 43400, Selangor, Malaysia; 2Department of Chemical and Environmental Engineering, Faculty of Engineering, Universiti Putra Malaysia, Serdang 43400, Selangor, Malaysia

**Keywords:** hydrogel, nanocellulose, processing methods, biomass, applications

## Abstract

Oil palm empty fruit bunch (OPEFB) is considered the cheapest natural fiber with good properties and exists abundantly in Malaysia. It has great potential as an alternative main raw material to substitute woody plants. On the other hand, the well-known polymeric hydrogel has gathered a lot of interest due to its three-dimensional (3D) cross-linked network with high porosity. However, some issues regarding its performance like poor interfacial connectivity and mechanical strength have been raised, hence nanocellulose has been introduced. In this review, the plantation of oil palm in Malaysia is discussed to show the potential of OPEFB as a nanocellulose material in hydrogel production. Nanocellulose can be categorized into three nano-structured celluloses, which differ in the processing method. The most popular nanocellulose hydrogel processing methods are included in this review. The 3D printing method is taking the lead in current hydrogel production due to its high complexity and the need for hygiene products. Some of the latest advanced applications are discussed to show the high commercialization potential of nanocellulose hydrogel products. The authors also considered the challenges and future direction of nanocellulose hydrogel. OPEFB has met the requirements of the marketplace and product value chains as nanocellulose raw materials in hydrogel applications.

## 1. Introduction

Lignocellulosic biomass material (LBM) is classified as a natural, non-toxic, abundant, sustainable, and renewable material. It is available either from woody or non-woody plants and is mainly composed of cellulose, hemicellulose, and lignin. More than 198 billion metric tons of LBMs are produced annually, and it has a great advantage as a cheap and highly available feedstock for numerous applications. Cellulose is the most abundant biopolymer that has a great potential to be developed in large scale commercial processes with a low selling price [1]. Recently, there has been a resurgence in demand to utilize cellulose for the production of advanced materials, which will undoubtedly become an important key in future bioeconomy [2]. Some common LBMs, also known as agriculture waste, have been used for producing high-end products such as jute, ramie, hemp, kenaf, bamboo, and oil palm.

Malaysia is a well-endowed country with oil palm and is the second largest producer after Indonesia, contributing more than 80% of the world production [3]. Oil palm is the most cultivated plant in Malaysia, with approximately 5.4 million hectares of plantation area and about 423 palm oil mills are operating in the country [4]. The huge plantation areas as well as tremendous amount of palm oil mills have caused Malaysia to be listed as the world’s top palm oil exporter for numerous palm oil products. In line with this, large biomass production occurs with around 90% from the remaining 10% oil extraction that come from two main sources: plantations and mills [5]. An enormous number of oil palm trunks and fronds are generated from plantations. Meanwhile, other biomasses like mesocarp fiber, kernel shell, and empty fruit bunches come from the milling process of fresh fruit bunches [6]. In Malaysia, oil palm empty fruit bunch (OPEFB) up to 22–23 million tons is abundantly generated as a residue annually [3]. As a non-wood fiber, OPEFB is considered as the cheapest natural fiber with good properties and has great potential as an alternative main raw material to substitute woody plants, which are expensive for various industries [7].

To date, biodegradable and environmentally friendly products have garnered worldwide attention including the whole processing line, especially in upfront sectors like automotive, textile, cosmetics, and packaging [8]. Due to insufficient petroleum resources and the high price of wood as the main raw material, LBM is a suitable alternative in producing regenerated cellulose products. Cellulose is a linear homopolymer composed of D-anhydroglucopyranose units (AGUs) that consist of β (1–4)-glycosidic bonds. Native cellulose which is also known as cellulose I, is a semi-crystalline polymer composed in a parallel arrangement and is not the most stable crystalline form [9]. Regenerated cellulose product, commonly known as cellulose II, has a similar molecular formula (C_6_H_10_O_5_)_n_ as cellulose I, but is more stable and can be molded into specific products such as membranes, hydrogels, aerogels, and fibers [10,11]. Cellulose II is the results of dissolution and recrystallization where the cellulose chains adopt an anti-parallel arrangement structure, which is the most stable form [12]. Regenerated cellulose is a material that is formed as a soluble cellulosic derivative including subsequent regeneration, typically in fiber, film, or gel-like forms, depending on their production methods and come from the conversion of natural cellulosic materials [11].

Hydrogels are usually made from petroleum-based synthetic polymers, either poly acryl amide (PAAM) or poly acrylic acid (PAA). These polymers are less environmentally friendly due to their non-degradability. Thus, extensive research on the modification of hydrogels has been undertaken by using natural polymers such as cellulose (polysaccharides), which is biodegradable, has good biocompatibility, is renewable, and non-toxic. However, they have inferior mechanical properties compared to petroleum-based hydrogels. Several techniques to improve the mechanical properties of hydrogels include the incorporation of suitable natural polymers or other inorganic materials [13]. In recent years, many nanomaterials have been introduced with promising great properties. These nanomaterials are usually known as cellulose nanocrystal (CNC or NCC), bacterial nanocellulose (BNC), and cellulose nanofiber (CNF). To cater for issues like poor interfacial connectivity and the mechanical strength of the produced hydrogel, nanocellulose is introduced due to its impressive properties such as lighter weight, higher surface area-to-volume ratio, and higher stiffness and strength compared to cellulose. It also has the ability to form effective hydrogen bonds within other polymeric matrices as well as across the cellulose chains [2]. Therefore, it is a promising material for use as a superior reinforcing material. The nanocellulose is incorporated as a reinforcing material into the polymer to form a cellulose fiber-reinforcement composite. CNF has micro-dimensions in length and nanodimensions in diameter compared to CNC, which has both a length and diameter in nanosize [14]. This review focuses on the CNF from OPEFB as a nanomaterial incorporated in hydrogel for various applications due to its ultralight and highly porous characteristics and is capable of being employed in various industries such as agriculture, biomedical, tissue engineering, food, and biocomposites [11,15]. CNF can be extracted via chemical or mechanical methods. The commonly used methods to produce CNF are explained in the next section. In particular, tailoring the swelling and mechanical properties of hydrogels to the needs of a specific application is essential to create versatile and high-performance functional materials. In addition, the formation of high modulus hydrogels (aside from specific composition combinations such as double network hydrogels) remains one of the barriers to the effective translation of hydrogels to practical applications. Furthermore, the apparent biological inertness of CNF makes them attractive in applications such as drug delivery systems, tissue engineering, scaffolds, and wound healing materials, all of which are particularly well-served by the high porosity, water content, and typical cytocompatibility of hydrogels [16].

However, the usage of petroleum-based materials has raised concerns regarding human health issues, greenhouse gas emissions, degradation time, depletion of energy resources, and endangered marine life. These issues have forced green technology development to use nature-friendly materials as an alternative for fossil-fuel based products [2]. Thus, this review briefly discusses upgraded hydrogels for various high-end applications which are produced from potential eco-friendly raw materials (such as OPEFB including versatile nanocellulose additives) with the hope that it could contribute to a better future bioeconomy. Furthermore, the upgraded hydrogels have the potential to replace current synthetic products that are harmful to humans and other ecosystems. In fact, the increase worldwide demand for hydrogel usage has been forecasted for the year 2024. Moreover, the mechanical strength and stability including the porosity of the nanocellulose hydrogel have been successfully enhanced [17]. However, extensive summaries focusing on the preparation of cellulose nanomaterial-based hydrogel/aerogel using OPEFB are still limited. Thus, this review highlights the potential usage of OPEFB as a main raw material for making hydrogels using recent several emerging methods such as 3D printing.

## 2. Oil Palm Empty Fruit Bunch (OPEFB)

Malaysia is endowed with a huge amount of biomass resources from (1) agricultural crops such as sugarcane, cassava, and corn; (2) agricultural residues such as rice straw, cassava, rhizome, and corncobs; (3) woody biomass such as fast-growing trees, wood waste from wood mill, and sawdust; (4) agro-industrial wastes such as rice husks from rice mills, molasses, and bagasse from sugar refineries and residues from palm oil mills; (5) municipal solid waste; and (6) animal manure and poultry litter [18].

Malaysia is one of the leading agricultural commodity producers in the Southeast Asian region. Therefore, agricultural wastes are abundant and readily available [19]. The main agri-based wastes that exist in the country are oil palm biomass (in the form of fronds, trunks and fibers), paddy straw, rice husk, banana residues, sugarcane bagasse, coconut husk, and pineapple waste [20]. The main agricultural wastes from the oil palm biomass accounted for 46,000 kilotons in the form of fronds and 11,000 kilotons in the form of trunks. In 2007, approximately 880 kilotons of paddy straw and 484 kilotons of rice husk were produced from the replanting of paddy [20]. Banana residues, sugarcane bagasse, and coconut husks accounted for 530, 234, and 171 kilotons, respectively. In pineapple farms, 48 kilotons of pineapple wastes are generated after the fruits are extracted [20].

Oil palm biomass is the major biomass resource in Malaysia. In 2017, due to the new planted area in Sarawak, the oil palm planted area in Malaysia reached 5.81 million hectares, with an increase of 1.2% from the 5.74 million hectares recorded in the previous year. The data of the oil palm planted area from 2013 to 2017 were obtained from Malaysian Palm Oil Board (MPOB) website, as shown in Figure 1. Oil palm biomass can generally be classified into oil palm fronds (OPF) and oil palm trunks (OPT), oil palm empty fruit bunches (OPEFB), palm kernel shells (PKS), mesocarp-fiber (MF), and palm oil mill effluent (POME). In total, 44.85 Mt of oil palm biomass is generated during the fresh fruit bunch processing, oil palm tree replanting, and pruning activities.

The OPEFB is a byproduct from the processing of crude palm oil (CPO) in a palm oil mill. It is obtained from the empty stalks of the fresh fruit bunches (FFB) after the fruits are separated from it. It is estimated that 4 kg of dry biomass is generated for every kilogram of extracted palm oil [22]. Therefore, approximately 22 to 23 million tons of OPEFB could be generated by the palm oil mills annually. The OPEFB has 50.9% cellulose, 29.6% hemicellulose, 17.84% lignin, 3.4% ash, and 3.21% extractive, making it a potential lignocellulosic material suitable for the production of bioalcohol, solid fuel, pulp, and many other value-added products [23]. Apart from its abundance, OPEFB has high cellulose, making it a very promising feedstock for the extraction of nanocellulose and the production of various cellulose-based products. Regenerated cellulose hydrogel, in particular, has been produced by dissolving OPEFB cellulose and sodium carboxymethylcellulose (NaCMC) in a sodium hydroxide/urea system [24]. Compared to other agricultural wastes, OPEFB is a great option as a potential raw material since it is composed of high cellulose content in addition to its abundant availability.

## 3. Regenerated Cellulose

### 3.1. Membrane

Most commercial regenerated cellulose membranes (RCMs) are prepared from cellulose due to its unique hydrophilic properties, high chemical stability, and ability to conserve the surrounding environment. Therefore, cellulose is an interesting raw material for an easy process development of RCM production [25]. Furthermore, cellulose membrane properties are mainly controlled by the surrounding environment, nature, and coagulation mechanism. Therefore, many efforts have been undertaken to investigate the coagulation or regeneration process to obtain a certain morphology or cellulose membrane properties [26,27,28]. In the past few years, a new and strong organic solvent, *N*-methylmorpholine-*N*-oxide (NMMO) has been developed, which can rapidly dissolve the cellulose without any complex formation [25]. Cellulose fibers and membranes prepared from the cellulose/NMMO/H_2_O solution show good mechanical and absorption properties when the cellulose solution agglomerated in water at low temperature via the phase changes method [25,29,30,31]. This is the conventional method to produce RCM without a filler. Furthermore, other methods are also available to incorporate the filler with the matrices via physical entanglement of polymer chains including the freeze-thaw method [13,32,33]. Nowadays, due to the advanced technology, the focus of the research world has mainly been on nanomaterials.

As is widely known, membranes are commonly used to separate different mixtures in a solution by allowing some particles to pass while the others are maintained. Two pivotal parameters for this purpose are selectivity and permeability. Thus, the efficiency of the membrane separation processes depends on these two factors. The introduction of a nanocellulose material with a combination of unique features showing chemical inertness, hydrophilic surface chemistry, high specific surface area, and high strength make it suitable as high-performance membranes. To fabricate membranes at various morphologies, several methods have been developed such as vacuum filtration [34], dip coating [35], electrospinning [36], and solvent casting [37]. Nanocellulose membrane is usually used as membrane materials for water purification because it can lower the organic fouling and biofouling, aside from virus removal, pollutant removal, bacteria removal, or separation of oil and water. Other than that, the polymer electrolyte with nanocellulose could not only reduce the water uptake, but also improve the thermal and mechanical stability including the reduction in volume and area swelling ratios in proton conducting membranes where it is a critical and pertinent material in both direct methanol fuel cells (DMFCs) as well as proton exchange membrane fuel cells (PEMFCs). Hence, the addition of nanocellulose could reduce the unwanted methanol crossover. In recent years, nanocellulose membranes for carbon dioxide (CO_2_) separation have attracted many researchers due to its promising performance, which have been reported followed by membrane distillation, organic solvent nanofiltration, and solar cells [38].

### 3.2. Hydrogel

Hydrogel is a three-dimensional cross-linked hydrophilic polymer with the ability to absorb a large amount of water, physiological, or saline solution. It is an attractive soft material, that is suitable and applied in many areas such as pharmaceuticals, food, agriculture, food packaging, electronics, drugs delivery, and personal care products. [39]. This is due to its hydrophilicity, permeability, good compatibility, and low coefficient of friction properties. Hydrogel can be divided into natural and synthetic hydrogels. Based on the basic crosslinking method, hydrogels can also be divided into chemical gels and physical gels. A physical gel is formed by the self-assembly of molecules through ionic and hydrogen bonding, whereas chemical gels are formed by covalent bonds [40].

Hydrogels based on synthetic polymers such as poly(vinyl alcohol), poly(amide-amine), poly(*N*-isopropylacrylamide), polyacrylamide, poly(acrylic acid), and their co-polymers have been reported to be formed by cross linking agents. Unlike natural hydrogels, synthetic hydrogels like polyethylene glycol (PEG)-based hydrogels with adjustable mechanical properties and an easily controlled chemical composition can be formed through the photopolymerization process. Many natural polymer-based hydrogels fabricated using hyaluronate [41], alginate [42], starch [43], and cellulose [44] have shown potential in the biomaterials area due to the hydrophilicity, biocompatibility, and biodegradation properties. Cellulose hydrogels can be prepared from a cellulose solution assisted by physical cross-linking since cellulose consists of a hydroxyl group, which can create a simple hydrogen bond network [44]. The existence of various hydroxyl groups in the cellulose molecule enables the crystalline formation to be bonded together by the hydrogen bond. There is an intra- and inter-molecule of the hydrogen bond with van der Waals forces that exist among the non-polar group in cellulose [45].

Hydrogels from natural polymers, especially polysaccharide, are suitable for application in biomaterials area because of its large quantity, non-toxicity, biodegradability, and biological use. Cellulose based hydrogels can be obtained from the cellulose derivative chemical cross-linked dissolved in the water by using small bifunctional molecules as a crosslink agent. Previous study has reported that a cellulose hydrogel was made by the direct dissolution process using aqueous NaOH/urea solution as a solvent and epichlorohydrin (ECH) as a cross-linker. From the study, the cellulose gelation behavior in the aqueous NaOH/urea solution and the effect of heating or cooling treatment on gelation production were investigated [44]. Since the first report regarding hydrogel application in the biomedical area, hydrogels have been used widely in biomaterial and pharmaceutical areas for various applications including tissue engineering and drug delivery due to the good compatibility. Nowadays, hydrogels are designed to react with the changes in the surrounding environment such as pH, temperature, and certain mixture. Hydrogels with responsive stimulation have huge potential in pharmaceutical areas due to the unique swelling and transparency properties that enable it to change into other physical, chemical, or biological stimulations [46]. Figure 2 shows the applications of cellulose hydrogels in many fields.

In recent decades, due to the capability of hydrogels to be used in a vast area and the emergence of new versatile materials such as cellulose nanomaterials have open up great new findings that show an excellent inherent chemical as well as physical properties like high specific surface area, high tensile strength, low density, high elastic modulus, reactive surfaces, and are renewable and biodegradable. These great properties have led to broad application prospects like electroconductives, biomedicals, and optical materials including reinforcing fillers. Due to a more uniform particle size distribution and higher specific surface area, a more mechanically stable self-assembled structure of hydrogel could be formed. Other than that, with the specific cross-linking strategy, nanocellulose hydrogels demonstrate a controllable morphology, high biodegradability, and biocompatibility as well as outstanding mechanical stability [17]. Compared to CNC, the CNF hydrogel is easily formed because it has more entanglement and flexibility. As such, many findings and published works on CNF hydrogels are available. In a previous study, Paako et al.’s (2007) research group was the first group to show that a CNF hydrogel was possible at low concentration, followed with enzymatic and homogenization treatments [47]. Meanwhile, a CNF hydrogel was successfully formed by adding salt or lowering the pH, and produced aligned hydrogels with oriented fibrils, which was further used as a template for anisotropic nanocomposites [48]. An alkaline treatment was also done on a pulp before defibrillation to alter the crystal structure of CNF because it consists of both crystalline cellulose I and amorphous cellulose [49]. The usage of NaOH concentration will affect the cellulose allomorph by the increase in cellulose II. The hydrogel with a cellulose II crystal structure showed an increase in Young’s modulus compared to the hydrogel with a cellulose I crystal structure due to the firm interdigitation of neighboring cellulose II nanofibers. However, most of the research studies have not reported on the crystallinity index or crystal structure of the CNF materials employed. Furthermore, CNF is frequently used as a reinforcing material to produce tough yet highly flexible hydrogels, especially in biomedical and tissue engineering applications [16].

The addition of cellulose nanomaterials in regenerated cellulose membranes and hydrogels could develop a new product with exceptional properties due to its own great features. This has been proven in several previous studies, which will be further discussed in the next section. As above-mentioned, the cellulose nanomaterials were classified into several categories: CNC, CNF, and BNC. The most common nanocellulose extracted from OPEFB are CNC and CNF.

## 4. Nanocellulose

Nanocellulose is a renewable nanomaterial with many potential applications in advanced materials, biomedical, and food packaging. It has outstanding properties like being lightweight, stiff, non-toxic, a high tensile index, and is most abundant on Earth. It can be derived from any resource material such as lignocellulosic fibers or so-called natural fibers that can be found in 2000 plant species [50]. Natural fibers consist of three main components: cellulose, lignin, and hemicellulose. Other components such as the extractives of polar and non-polar components can also be extracted from natural fibers. In other words, the natural fibers are composed of cellulose microfibrils that are structured in a matrix together with lignin and hemicelluloses components [51]. Natural fibers depend very much on the cellulose type that is related to crystalline composition. The mechanical properties of natural fibers are influenced by the organization of crystalline composition [52]. A single cell of natural plant fibers has a 1–50 mm length and 10–50 m diameter that are formed from the cellulose microfibrils. The microfibrils are formed of 30–100 cellulose molecules with a diameter of 10–30 nm. Many types of extraction methods are applied to isolate the fibers from the natural plant stem.

Microcrystalline cellulose (MCC) and cellulose nanocrystals (CNC) are two types of common cellulose with a particle size diameter that ranges from 10 to 50 μm [53] and 3 to 5 nm, respectively. In addition, MCC is isolated via sulfuric acid (H_2_SO_4_) treatment, while CNC is isolated via acid hydrolysis. In terms of structure, MCC has crystalline structures of multi-sized cellulose microfibril aggregates that appear in bundles [54], while NCC exists as whiskers or known as rod-shaped in crystalline regions [55].

Nanocellulose can be categorized into three nano-structured cellulose: (i) CNF, (ii) CNC, and (iii) BNC. These nanocelluloses differ based on the method of production. To date, huge numbers of research have used OPEFB as the raw material to produce CNF and CNC. However, study on bacterial nanocellulose, which is made of OPEFB cellulose nanofibrils that involve bacteria or enzymatic polymerization is not available. The CNF and CNC can be self-prepared via chemical, mechanical, or combination methods. Most studies have incorporated nanocellulose in the manufacture of composites, paper, or film.

To address the issue of low mechanical strength, absorbability, or transparency of the regenerated cellulose membrane/hydrogel, the incorporation of nanocellulose is very practical. The mechanical strength is successfully enhanced by employing nanocellulose as the additive. These improvements can be applied for high-end applications such as for water filtration, mulching mat, wound healing patch, and other applicable products. The increase in some properties like mechanical strength will have a significant impact on the usage of the products. Unnecessary pretreatments or processes that will increase the production cost could be eliminated. For instance, in the papermaking process, the addition of nanocellulose can reduce the beating revolution which will indirectly reduce the energy consumption and production costs. At the same time, a better fiber bond can still be achieved.

### 4.1. Cellulose Nanofibers (CNF) from OPEFB

The CNF prepared by hydrolyzing OPEFB with sulfuric acid [56] has an average width of 1–3.5 nm by varying the time of hydrolysis. A longer period of hydrolysis produces nanofibers with better yield, lower degree of polymerization, and crystallinity. The CNF extracted from OPEFB using chemo-mechanical processes such as H_2_SO_4_ hydrolysis and high-pressure homogenization produced CNF with sizes of 5 to 10 nm [57]. CNF can also be extracted from OPEFB through the ultrasound effect during the stages [58] and can be obtained after the soda-anthraquinone pulping and bleaching processes. An ultrasound equipped with the frequency of 20 kHz and the output power of 700 W was used to produce CNF with a diameter of 5 to 23 nm. MFCs can be prepared using two different techniques: ammonium persulfate oxidation and sulfuric acid hydrolysis [59]. Both techniques produced MFCs of long and network-like fibrils with widths ranging from 8 to 40 nm. One study isolated CNF from OPEFB via the thermal-chemical process followed by nano-grinding treatment [60]. The produced nanocellulose had a morphological dimensional change from 8.25 μm to 17.85 nm. Another study also used the isolation of OPEFB to produce lignocellulose nanofibers (LCNFs) [61] by applying multi mechanical stages with varied vibration milling times. The external surface of the produced nanofibers was uneven, irregular, folding, and unsmooth, with an optimal size of 53.72–446.80 nm.

The cellulose OPEFB fiber can be converted into MFC and CNF through peracetic acid delignification followed by enzyme hydrolysis [62]. The enzyme hydrolysis can be used as a method to transform cellulose into MFC, but it does not have the capability to become a nanocellulose. Hastuti et al. (2019) characterized CNF derived from OPEFB that could produce by 2,2,6,6-tetramethylpiperidine 1-oxyl (TEMPO)/NaBr/NaClO [63]. The crystallinity indices (CrIs) were identified to range from 34% to 55% by using x-ray diffraction (XRD). High-performance nanomaterials like TEMPO-oxidized CNF were successfully prepared with good characteristics from low-quality biomass waste such as OPEFB. Other recent work prepared nanocelluloses from chemically purified celluloses of oil palm empty fruit bunch (CPC-OPEFB) by using acid hydrolysis [64]. The nanocelluloses from CPC-OPEFB were prepared with sulfuric acid treatment at the concentration of 67 wt% at 40 ± 1 °C for 10, 20, 30, and 40 min. The particle size analysis proved that the diameter of the obtained nanocelluloses was affected by the hydrolysis time. The best hydrolysis time to obtain the smallest diameter of CNFs from CPC-OPEFB was 30 min.

In the making of film, Lani et al. (2014) prepared nanocellulose from OPEFB fiber that had a 4 to 15 nm diameter [65]. The nanocellulose was applied to reinforce the polyvinyl alcohol/starch blend films, and 5% (v/v) of nanocellulose formulated the best nanocomposites with the tensile strength of 5.694 MPa. Salehudin et al. (2014) incorporated nanocellulose extracted from OPEFB to enhance the mechanical properties of starch-based polymers [66]. The CNF was prepared by hydrolyzing OPEFB with 64% H_2_SO_4_ at 45 °C for 90 min and obtained nanofibers with diameters of 50 to 90 nm. The incorporation of 2% nanofiber enhanced the starch-based film up to 28% in terms of tensile strength. Another study [67] found that 1 wt% of OPEFB nanocellulose reinforced poly(vinyl alcohol)-α-chitin composite films were improved by 57.64% and 50.66% of tensile strength and Young’s modulus, respectively.

Nanocellulose is also used in preparing nanopaper. Ferrer et al. (2012) prepared dissimilar cellulose pulps of sulfur-free chemical treatments of OPEFB [68]. The pulps were microfluidized to obtain CNF, which was used to manufacture nanopaper via an overpressure device. The nanopaper had lesser water absorption, higher tensile strengths (107–137 MPa), and higher elastic modulus (12–18 GPa).

A superadsorbent was produced from OPEFB for water remediation through sulfuric and phosphoric acid hydrolysis with activated carbon [69]. The occurrence of sulfonic groups achieved better remediation capabilities on the NCS compared to NCP. The performance was doubled compared to the sample of rice-straw NC by having a metal adsorption capability to Pb^2+^ with 86% efficiency and 24.94 mg/g adsorption capacity.

### 4.2. Cellulose Nanocrystal (CNC) from OPEFB

A study successfully isolated MCC from OPOPEFB-total chlorine free pulp [70]. The acid hydrolysis method was applied using the TCF pulp bleaching that was done via the oxygen–ozone–hydrogen peroxide bleaching sequence. The produced MCC had 87% of crystallinity, which had good thermal stability. CNC was also isolated from OPEFB-total chlorine free bleached pulp by the acid hydrolysis of 58% sulfuric acid concentration continued by ultrasonic treatment [71]. The optimal hydrolysis time was 80 min for CNC with dimensions of 150 nm in length and 6.5 nm in diameter. It was proven that the CNC could be practically produced from chlorine free pulp, which are known as environmentally benign processes as they save energy and reduce chemical usage. Pujiasih et al. (2018) focused on the silylation of MCC from OPEFB via the aminosilane compound synthesized through the aminolysis of 3-glycidoxypropyltrimethoxysilane with ethylenediamine [72]. Three steps were involved: (i) bleaching process, (ii) alkaline treatment, and (iii) acid hydrolysis. Budhi et al. (2018) used OPEFB as raw material to obtain CNC [73]. They managed to produce 44.8% yield of CNC from dried OPEFB with a diameter about 140 nm and crystallinity index at 73.3%. Septevani et al. (2019) synthesized and characterized nanocellulose obtained from OPEFB via strong H_2_SO_4_ and mild acid (H_3_PO_4_) hydrolysis at 50 °C for 3.5 h. A rod-like and long filament-shaped nanocellulose was obtained from the strong and mild acid hydrolysis, respectively. The degree of crystallinity was higher from the strong acid hydrolysis (96%), compared to that of the mild acid hydrolysis (86%).

An initiative to reduce the agglomeration problem was carried out and the CNC from OPEFB was prepared using the TEMPO (2,2,6,6-Tetramethylpiperidinyloxy or 2,2,6,6-Tetramethylpiperidine 1-oxyl)-oxidation reaction method [74]. The drying and solvent exchanged techniques were applied in the post-treatment step, and the agglomeration of NCC due to the hydrogen bonding among cellulose linkages was minimized.

Another study showed that the application of 1–10% CNC as a reinforcement material in the composite exhibited good transparency and visibility by obtaining more than 80% of visible light transparency at 550 nm [75]. The CNC from OPEFB was produced by using chemical pulping such as soda pulping, followed by the 2,2,6,6-tetramethylpiperidine-1-oxy (TEMPO) oxidation reaction method. The produced NCC was used to enhance the polylactic acid (PLA) biopolymer film matrix by 0–20% of loadings. The NCC had a rod-like shape of 2–6 nm in width and 200–500 nm in length.

## 5. Potential Preparation Methods for the Oil Palm Empty Fruit Bunch Nanocellulose Hydrogel

Nanocellulose is a natural material extracted from plant cell walls at a nanometer scale size. It is hydrolyzed by mechanical, enzymatic, or chemical treatment to remove lignin and hemicellulose components and the cellulose component is reconstructed into partially to fully crystalline nanoparticles [76]. The high aspect ratio feature of nanocellulose contributes properties such as promising strength, high stiffness, and high biocompatibility, therefore, it has become a popular reinforcement material in many advanced application fields such as nanocomposites, biomedicals, automotive, and construction. A variety of nanocellulose in terms of length, diameter, and surface charge can be found by different preparation methods, depending on the end usage. Although the study on oil palm empty fruit bunch (OPEFB) nanocellulose application in hydrogels is limited, many investigations on OPEFB nanocellulose fabrication are available, as mentioned in the previous section. Therefore, there is no doubt that there is a high potential to prepare nanocellulose hydrogels from OPEFB by using the fabrication methods that prepared the current nanocellulose hydrogel.

Nanocellulose hydrogel is a highly porous lightweight material with an average pore size in the tens of nanometers. The OPEFB nanocellulose can be available from 1 to 3.5 nm by varying the time of hydrolysis [56]. The wide range of processing methods for preparing nanocellulose hydrogels also adds to the broad interest and application of these materials. High stiffness nanocellulose is able to provide enhanced performance and stability to the hydrogel, even in low portions due to the high number of hydrogen bonds and entanglement of fibrils [77]. The insertion of 2 wt% of OPEFB gave a higher stiffness of specimens [78]. The insertion of nanocellulose enhanced the modulus and compression strength due to the good dispersion of nanocellulose in the hydrogel, and hence more effective crosslinking density and interfacial adhesion [77]. Figure 3 shows that the nanocellulose hydrogel had a smaller pore size, which is because the nanocellulose fibers were distributed evenly inside the three-dimensional (3D) porous material, making it lower in swelling capacity [79]. The hydrogel was first frozen by using liquid nitrogen and lyophilized for 24 h at 55 °C. Then, the hydrogel was deposited directly onto aluminum stubs using two-sided adhesive carbon tabs for scanning electron microscope viewing. A similar scenario was also reported in the OPEFB nanocellulose film [80].

To produce an OPEFB nanocellulose hydrogel, numerous potential processing methods can be used such as homogenization, grafting of OPEFB nanocellulose in the polymerization process, freeze-thawing, and 3D printing technology. Nonetheless, OPEFB nanocellulose hydrogels will have a huge impact in many fields.

### 5.1. Homogenization Processing Method for Nanocellulose Hydrogel

In the homogenization method, usually known as a simple mixing method, the suspension materials (nanocellulose and hydrogel polymer) are simply put into a container with organic acid and stirred overnight at room temperature. After that, the hydrogel is washed with NaOH to remove all acid and residual NaCl. To obtain crosslinked nanocomposites, the hydrogel is soaked in a cross-linker solution. Figure 4 shows the schematic diagram of the nanocellulose collagen hydrogen preparation. A previous study reported that not all cross-linker agents are suitable for biomedical applications. Figure 5 shows that the cross-linked hydrogel had a relatively lower cell growth and adhesion compared to the nanocellulose hydrogel without cross-linking [81]. A production process that works without heat should minimize the damage in the hydrogel due to the lower cross-linking profile at higher temperature. One study showed that the viscosity of the nanocellulose hydrogel was reduced by about 350%, when the temperature increased from 5 to 30 °C, which made the hydrogel difficult to model [82]. Furthermore, this method does not require high skills to fabricate the hydrogel.

### 5.2. Grafting of Nanocellulose Hydrogel—Surface-Initiated Free Radical Polymerization

One of the most commonly used approaches to graft vinyl polymers from the surface of CNCs is surface-initiated free radical polymerization [84]. The incompatibility between hydrophilic nanocellulose with the hydrophobic polymer matrix creates severe issues like filler aggregation and low interfacial bonding can lead to poorer hydrogel performances. Hence, nanocellulose can be bonded with grafted polymer with similar surface characteristics and other physicochemical properties through the fiber grafting process. The nanocellulose is grafted with the monomer, followed by the polymerization process to produce nanocellulose hydrogel. The persulfate surface initiator (as well as the radical initiator in the FRP process) allows radical formation to start in the aqueous phase at temperatures ranging between 60 and 70 °C. The free sulfate radicals thus produced are capable of abstracting hydrogen atoms from the surface of the nanocellulose and thus generate the surface-bound initiating species (Figure 6). However, the CNF grafted monomers in the FRP method will not be involved in radical transfer because they do not have vinyl group reactive sides.

FRP is a method of continuous polymer chain building by using free radicals. Therefore, it is a rapid process that can be carried out as a bulk treatment, but is insensitive to impurities. The FRP process can be kickstarted by separating the initiator molecules, followed by the radical reaction with the monomer. The radical decay can be initiated photochemically, electrochemically, chemically, or thermally [85]. Thermal initiator persulfates like potassium persulfate, ammonium persulfate [86], and sodium persulfate [79] are the most widely used in previous studies, as listed in Table 1. Denisov (2003) wrote a handbook of free radical initiators and explained the kinetic and mechanism of radical decay in detail [87].

After that, the radical monomer attacks the neighboring monomers, resulting in a longer polymer chain until the two radical polymer chains are combined. This will terminate the chain growing process by the deactivation of both radicals.

Atom transfer radical polymerization (ATRP) is a controlled/living radical polymerization method that can also produce nanocellulose hydrogels. Unlike FRP, it uses the application of reversible metal-catalyzed atom transfer to generate the propagating radicals. This process is initiated by the initiator. It determines the dormant chain structure and is able to combine with more than one monomer unit. The coupling or disproportionation reduces the polymerization process to less than a few percent, but the process never terminates in ATRP. Mozafari et al. (2014) reviewed the biomaterials including hydrogel scaffolds by the ATRP processing method [88].

Ring opening polymerization (ROP) is a well-established technique used to polymerize cyclic monomers such as lactides and lactones [89]. The initiator is often equipped with alcohol functional group for the polymer’s ring opening [90]. ROP operates through different mechanisms, depending on the types of catalytic system, initiator, or monomer. Effectively, by tuning the alcohol-to-monomer ratio, the molecular weight of the final polymer can be controlled [91].

**Table 1 materials-13-01245-t001:** Details of previous studies using the free radical polymerization processing method to produce nanocellulose hydrogel.

Polymer/Monomer	Source of Nanocellulose	Water Source	Initiator, Cross-linker, Catalyst	Ref
Acrylamide (AA)	Cotton pulp	De-ionized (DI) water	Ammonium persulfate (APS), *N*,*N*’-Methylene-bisacrylamide (MBA), *N,N,N*′,*N*′-tetramethylrthylenediamine (TMEDA)	[92]
AA	Commercial bleached kraft soft wood pulp	DI water	Potassium persulfate (K_2_S_2_O_8_), MBA, TEMED	[93]
AA	Cotton fibers	Distilled water	K_2_S_2_O_8,_ MBA, TEMED	[94]
Sodium acrylate (SA)	Cellulose pulp	Double distilled water	K_2_S_2_O_8,_ MBA, TEMED	[95,96]
Cassava starch and SA	Cotton fibers	distilled water	K_2_S_2_O_8,_ MBA, -	[97]
N-Isopropyl acrylamide (purified by recrystallization iꞑ-heptane twice)	Acrylate-functional nanocellulose & HCl nanocellulose	Unclear source of water	APS_,_ MBA, TEMED	[98]
AA	Polar wood	Unclear source of water	K_2_S_2_O_8_, MBA, -	[99]
N-isopropyl acrylamide	Bleached bamboo pulp	DI water	K_2_S_2_O_8,_ MBA, *N*,*N*,*N*’,*N*’-tetramethyl-ethane-1,2-diamine (TMEDA)	[100]
Softwood fibers (Norwegian spruce)	Plant-derived nanocellulose	Ultrapure water	Not mentioning	[101]
AA	Bleached wood pulp	DI water	K_2_S_2_O_8,_ MBA, - Ionic crosslinker (CaCl_2_)	[102]
2-Dimethylamino ethyl methacrylate	Spruce bleached soft wood pulp	DI water	APS, MBA	[103]
AA	Softwood kraft pulp	DI water	APS, MBA, TEMED Ionic crosslinker (Iron (III) chloride hexahydrate [FeCl_3_·6H_2_O])	[104]

### 5.3. Freeze-Thaw Cycle Processing Method for Nanocellulose Hydrogel

The grafting process is not favored by many perspectives, especially biomedical related products, due to the involvement of many chemicals. Freeze-thaw processing is one of the optional methods in producing hydrogels without any chemical cross-linkers [105]. This method is experimentally straight forward and no chemical crosslinking agents are needed. Polymers that have a high molecular weight, degree of hydrolysis, and concentration are the most suitable for hydrogel fabrication via the freeze-thawing approach, and poly(vinyl alcohol) (PVA) is one of the most frequently used polymers [106].

The gelation of polymers in this processing method is controlled by the phase separation, which occurs as the solution freezes and the polymer is rejected from the growing ice crystallites [107]. This product is then refined with repeated cycling. The size of the ice crystallites increases with cycling, and the resultant gels are composed of water-filled pores where the ice has melted and is surrounded by a polymer skeleton. However, the CNF association affects the ice nucleation rate [108]. CNF forms thin films templated around ice crystals, and the repetitive freeze–thaw cycle offers a reconstruction of the CNF structure, resulting in more extensively assembled and thicker films as well as less unassembled thin fibers. However, the filler content is limited at a 2 wt% [108], instability hydrogel formation for higher nanocellulose contents [109]. Hydrogen bonding and polymer crystalline can further straighten the hydrogel via physical crosslinking. Many studies have reported that the concentration of the polymer solution, polymer molecular weight, freezing–thaw period, and the number of cycles affect the cross-linking degree of the hydrogel [110,111,112,113,114]. On the other hand, nanocellulose hydrogen is often used to enhance the mechanical properties of hydrogel [115]. Mechanical stirring, followed by ultrasonic stirring in distilled water, is conducted to avoid agglomeration in nanocellulose. Then, the polymer is inserted into the nanocellulose-dispersed solution, heated with a magnetic stirrer to dissolve the polymer homogenously with nanocellulose. Finally, the nanocellulose polymer mixture undergoes the freeze–thaw cycle to produce hydrogel. Figure 7 shows the simple process for nanocellulose hydrogel production.

### 5.4. 3D Printing Nanocellulose Hydrogel Processing Method

In the current industrial revolution (IR4.0), additive manufacturing represents a major innovation in manufacturing. 3D printing technology requires fewer labor and energy, and produces less waste but produces highly complex geometry [116]. The advantages of 3D printed hydrogels were reviewed by Gatenholm (2016) [117]. However, not every 3D processing method is suitable for hydrogel production. Liu et al. (2020) and Jang et al. (2018) reviewed the most frequent used hydrogel processing method in detail and its limitations are shown in Table 2 [118,119]. Although in 3D printing, no study on OPEFB nanocellulose has been conducted, the proven strength enhancements by OPEFB nanocellulose have high potential uses in 3D printing to produce OPEFB nanocellulose hydrogels.

Nozzle-based 3D printing is the most popular method to produce hydrogel scaffolds. The viscous hydrogel is placed into a syringe-like medium and extruded onto a building bed layer-by-layer to solidify. This method allows good hydrogel interlayer adhesion, but is limited in being well-known in various parameters. However, this driver system is not preferred in bio-applications due to the large pressure drops at the nozzle, which may harm the cells.

One of the biggest advantages of the inkjet 3D printing method is that a relatively small printing dimension is achievable such as soft tissue applications where minimum contamination is required [121]. A piezoelectrical actuator or heat breaks the hydrogel into droplets (15 to several hundred microns) and the droplet falls on the building bed by gravitational pulls. However, a large and complex hydrogel product is not preferred and a piezoelectric actuator or electrical heating may be used to create pressure to eject the droplets [120].

Laser-based 3D printing method uses laser energy, usually in the UV range, to build a 3D product in a pool of photocurable hydrogels. The exposure of the laser on the photocurable hydrogel forms a thin gel-layer with a specific structure according to the pre-set in the computer system. Then, the gel-layer is stacked layer-by-layer to produce a 3D structure product with a high accuracy of dimension production and good quality of surface finishes under the optimum parameters (laser wavelength, power, spot size, scanning speed and exposure time) [122,123]. This process system can build the product in a huge scale, provided that the laser is reachable.

The OPEFB nanocellulose hydrogel is a potential solution to improving the mechanical strength as well as bioactivity [124]. The insertion of nanocellulose can improve the printability of the hydrogel component. Nanocellulose has a high viscosity at low shear rates and is shear thinning when extruded from a 3D printer. The high viscosity gives excellent shape fidelity [125]. The OPEFB nanocellulose synthesized by the TEMPO-oxidation reaction had high viscosity and performed gel-like structure [74]. The nanocellulose contents improved the mechanical properties of the specimens, tuning its strength from 3 to 8 kPa with different printing angles, thus showing a high compatibility with 3D printing technology in biomedical products. The enhancement effects may be contributed by the OPEFB nanocellulose. Furthermore, Leppiniemi et al. (2017) and Athukoralalage et al. (2019) investigated and reviewed 3D printed nanocellulose hydrogels in advanced applications, respectively [126,127]. These studies showed that the OPEFB nanocellulose hydrogel has the capability to be used in advanced applications.

Various potential processing methods can be utilized such as homogenization, grafting of OPEFB nanocellulose on polymerization process, freeze-thawing, and current 3D printing technology. However, each method has different approaches, thus, it depends on the complexity of the product development. Despite the simplicity of the homogenization process and no extra handling skills are needed, it makes the hydrogel difficult to model, and this will not be applicable for certain areas like the biomedical sector. Similar to the homogenization method, the grafting method is also not really suitable in the biomedical field due to the high chemical usage. Furthermore, it is not sensitive to impurities, despite its rapid process, which can be carried out as bulk. Last but not least, the 3D printing method comes with advanced technology and can improve many of the hydrogel properties, but still has several shortcomings that require further research. Since OPEFB is a potential raw material that has not been fully utilized yet with these processing methods, it is a new challenge and finding in the hydrogel-making area.

## 6. Potential Applications of OPEFB Nanocellulose Hydrogel

Nanocellulose hydrogel has versatile applications in various fields such as agriculture, food, energy, textiles, biomedical, and biocomposites. It has its own advantages when compared to the hydrogels made of non-degradable and toxic materials in the agricultural fields. Furthermore, nanocellulose hydrogels are friendly toward the soil pH as it would not drastically change the pH value of the soil [128]. Due to its capacity to retain a large amount of water, reported as 200 times that of its mass, nanocellulose hydrogels are favored in the agricultural sector [129]. Nanocellulose hydrogels could be used in biomedical applications including drug delivery, wound dressing, bioimaging, and wearable sensors [130]. In drug delivery systems, the application of nanocellulose hydrogels could improve the transport of therapeutic agents to the targeted tissues and organs. It also has shown potential in the application for tissue engineering. In a brief review compiled by Athukoralalage et al. (2019), the application of 3D bioprinting nanocellulose hydrogels prepared using 3D bioprinting for tissue engineering was discussed [127]. The 3D printed structures enhanced the precision and resolution. Tissue engineering aims to regenerate the injured tissues to reconstitute bone/cartilage and heal wounds in a less painful manner in shorter times.

Other than food itself, nanocellulose hydrogels have also been utilized as intelligent packaging as it is able to detect CO_2_ because the hydrogel consists of large amounts of water and offers fast proton generation in response to external CO_2_ stimuli. In fact, CO_2_ sensitive indicators are dependent on the pH change when CO_2_ dissolves in water, where the water plays an important role in the detection of CO_2_. Thus, nanocellulose hydrogels have great potential as a smart packaging material. Lu et al. (2018) showed a CNF hydrogel indicator that was exposed to fresh-cut fruits [131]. The study showed that the changes in color from dark green, which indicated a still fresh fruit, to an orange yellow that indicated that it had already spoiled after two days, as shown in Figure 8. This remarkable change means that the fruit was already contaminated with microorganisms that generate high CO_2_ content. The efficiency of the CNF hydrogel in the absorption of CO_2_ in its water molecules has the ability to quickly generate protons and give a fast response time. Thus, fruit spoilage can be easily detected.

Although the usage of hydrogel in various fields has already been explored, very limited studies on hydrogels made from OPEFB nanocellulose have been reported. Nevertheless, the potential applications of the OPEFB nanocellulose hydrogels could be anticipated based on studies that have reported on cellulose-based hydrogels. Table 3 displays the hydrogel made from OPEFB cellulose and its potential applications. For instance, Xiang et al. (2016) examined the feasibility of MCC extracted from OPEFB itself and its stalks and spikelet to be applied for food applications. It was reported that the cellulose content was the highest in the stalk fibers of OPEFB, while the spikelet fibers of OPEFB yielded the lowest content of cellulose. Correspondingly, the stalk fibers also possessed the lowest lignin and residual oil content. On the other hand, the crystallinity index of the spikelet fiber MCC was the highest compared to that of the stalk fiber MCC and OPEFB MCC, indicating that it is a suitable candidate in load bearing applications such as biocomposites [132]. Hydrogels made from OPEFB stalk fiber MCC were proven to be comparable with the commercial MCC and have the potential to substitute commercial MCC hydrogels in food and pharmaceutical products.

Salleh et al. (2018) produced regenerated superabsorbent hydrogel by dissolving cellulose into a mixture of NaOH/urea solvent and NaCMC [134]. Superabsorbent hydrogels possessing the swelling ability of more than 100,000% were produced. The fabricated hydrogel could be adopted in the tissue engineering technology. Apart from that, superabsorbent hydrogel fabricated by Salleh et al. (2019) using OPEFB cellulose was reported to have a swelling ability of more than 80,000% [24]. This kind of hydrogel could provide constant watering to plants. Gan et al. (2018) produced an aerogel from OPEFB cellulose and graphene oxide and the fabricated aerogel exhibited a macroporous structure with an equilibrium-swelling ratio ranging from 2000 and 3700% [135]. Due to its superior thermal stability, it has the potential to be used as a thermal insulating material. The aforementioned studies reveal the potential of OPEFB cellulose in the fabrication of superabsorbent hydrogels. Therefore, even if the hydrogel made from OPEFB nanocellulose has yet to be reported, it could be deduced that the OPEFB nanocellulose has the same potential or even more promising prospects in the fabrication of superabsorbent hydrogels to be applied in various fields.

## 7. Challenges and Future Directions

As the second largest producer of palm oil in the world, Malaysia is blessed with abundant oil palm biomass that could be beneficial as the feedstock for biochemical and biofuel synthesis. OPEFB is among the solid waste biomass that exists abundantly in palm oil mills. Around 22 to 23 million tons of OPEFB could be generated annually. However, only 10% of them are used, while the remaining are discarded. OPEFB has high amounts of cellulose, which makes it a very suitable feedstock for the extraction of nanocellulose. Due to its characteristics such as low cost and density, superior specific strength, and thermal stability as well as biodegradability, nanocellulose extracted from OPEFB has recently garnered a lot of attention from researchers. However, the application of nanocellulose extracted from OPEFB has yet to be widely explored, particularly in hydrogels. Nevertheless, the future prospect of this nanocellulose is very promising.

Although intensive studies have been conducted on nanocellulose hydrogels from other resources, there are still some research gaps that need to be filled. A comprehensive and more fundamental knowledge on the full interaction mechanisms of nanocellulose with biomolecules are necessary as well as its long-term effects to the human body [136]. Therefore, despite its advantages, nanocellulose hydrogels still face various challenges on the road to commercialization. One of the challenges is the long-term biosafety of the nanocellulose hydrogel in various applications, especially biomedical. Although nanocellulose hydrogels are generally considered as green and environmentally friendly, information on their biological impacts and life cycle is lacking. Studies on the life-cycle assessment (LCA) of hydrogels is very limited, let alone nanocellulose hydrogels.

To the best of our knowledge, the only LCA study on hydrogels was reported by De Marco et al. (2016), who studied the LCA of starch aerogels for biomedical applications [137]. The authors reported the relative contributions of three characteristic steps of aerogel production on each impact category such as human health, ecosystem quality, climate change, and resources. The mentioned three steps of aerogel production consist of: (1) gelatinization with the formation of hydrogel; (2) alcogel formation; and (3) supercritical drying to obtain the aerogel. Based on that study, the third step in aerogel production, which is the supercritical drying process, contributed the highest impact to all the studied categories, with the exception of respiratory organics. It could be attributed to the high energy consumption where supercritical CO_2_ was used to dry the structures.

On the other hand, respiratory organics that could have negative impacts on human health were reportedly emitted during the second step, where ethanol was used to substitute water in the hydrogel during the formation of alcogel. The authors recommended using shorter drying times and lower amounts of CO_2_ to reduce the energy consumption. Unfortunately, this is the only LCA study on the hydrogel and therefore comprehensive information on the topic is still scarce. Such information is vital for the future determination of biocompatibility and the hazard assessment of nanocellulose hydrogels. The life cycle of nanocellulose hydrogels is much more complicated when compared to the aforementioned study. It involves the processes of biomass production, and the separation of cellulose from other compounds present in biomass such as lignin and hemicelluloses, solvent production, cross-linking, and nanocellulose, hydrogel production, consumption, and biodegradation. Every procedure has to be included in hydrogel LCA studies to assist in identifying alternative resources and feasible procedures that will result in lower impacts to both the environment and humans [138].

Another challenge faced by nanocellulose hydrogels is the high cost and energy consumption in the production of both nanocellulose and hydrogel. These are the main factors that inhibit most developing countries in adopting these technologies widely in their nation. Therefore, it is recommended that future works should emphasize seeking a more cost-effective synthesis route to enhance the competitiveness of nanocellulose hydrogels on the market. Apart from that, more future work should also be conducted to find a more environmentally friendly approach in producing nanocellulose hydrogels [139]. Finally, exposure of the researchers to the requirements of the marketplace and product value chain is an important future topic so that the research in the lab could be well-fitted to industrial-scale processes.

## 8. Conclusions

This review describes the regenerated cellulose hydrogel from OPEFB for various applications that acquire additive materials with advanced properties. OPEFB resources can be considered as the cheapest raw material compared to other commercialized woody plants. It is abundantly available in Malaysia and has comparable properties to other woody plants. Despite it being classified as an agricultural waste, OPEFB is able to produce high end-products with great properties like hydrogels. Many studies have been conducted on nanocellulose production from OPEFB. Therefore, the potential is there for nanocellulose hydrogels. Nanocellulose hydrogels can be produced in various desired shapes, depending on its application. With the incorporation of nanomaterial, many hydrogel properties can be boosted to higher levels. Many favorable properties of nanocellulose hydrogels can be highlighted in this review such as ultralight, biodegradable, hydrophilic, biocompatible, cost-effective, environmentally friendly, high mechanical strength as well as excellent inherent physical and chemical properties. These great properties assist in the development and improvement of many sectors including food, agriculture, biomedical, tissue engineering, biocomposites, and several other areas. The emergence of nanocellulose hydrogels could be one of the best alternatives to replace petroleum-based related products, which have numerous uses in vast areas where it is not sufficiently environmentally focused. Moreover, petroleum-based materials contribute to a great number of global issues in terms of human health, pollution, non-biodegradability, and the depletion of energy resources. Thus, this new nanocellulose hydrogel technology creates new insights for more advanced green materials with superior properties. Furthermore, the raw material used would be from agricultural biomass (OPEFB), which is abundantly available with a low density and cost, high specific strength, and thermal stability as well as biodegradability. This review also emphasized several methods that have been used to produce nanocellulose hydrogel with the recent technology, 3D printing, particularly taking care of the hygiene and complexity of the products.

The production of OPEFB nanocellulose has been previously studied, but the information on OPEFB nanocellulose hydrogels is still limited. Moving forward, there are still some challenges and gap in this research needed to be filled. Even though the great properties of nanocellulose hydrogels have been discovered by many researchers, there are some parts that need to be understood and improved, especially when the produced materials are used for the human body. Related applications such as food, biomedical, and tissue engineering have particular concerns regarding the long-term effects, LCA, and health as well as market acceptance. In future research, these raised issues can be catered to achieve the feasibility of these products. With continuous research into nanocellulose hydrogels, the properties of the products will be improvised, and the development of prospects will be further explored. In a nutshell, the nanocellulose hydrogel is a strong potential candidate for numerous high-end applications in different industries. The surfacing challenges are expected to be overcome in the near future. Its exceptional properties will lead to “green” cost-effective products that will provide significant advances and improvement in people’s quality of life.

## Figures and Tables

**Figure 1 materials-13-01245-f001:**
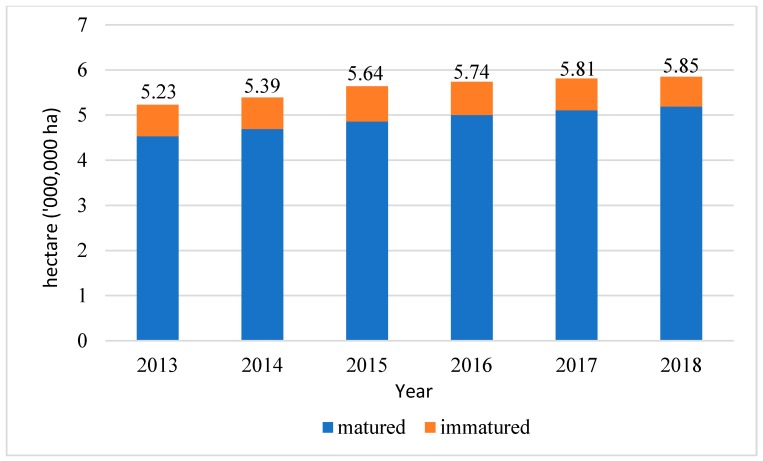
Oil palm planted area in Malaysia from 2013 to 2018. The figure above was adapted with permission from the cited reference [21].

**Figure 2 materials-13-01245-f002:**
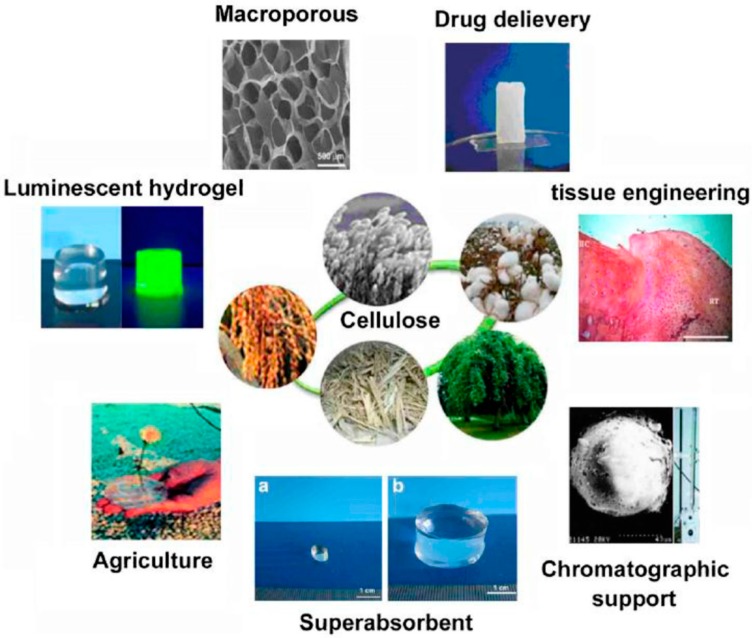
Application and development of cellulose-based hydrogel. The figure above was adapted with permission from the cited reference [44].

**Figure 3 materials-13-01245-f003:**
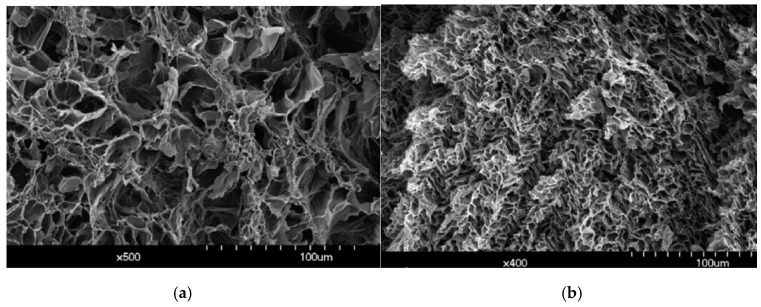
Scanning electron microscope micrographs of (**a**) a polyacrylamide polymer hydrogel and (**b**) nanocellulose content hydrogel. The figure above was adapted with permission from the cited reference [79].

**Figure 4 materials-13-01245-f004:**
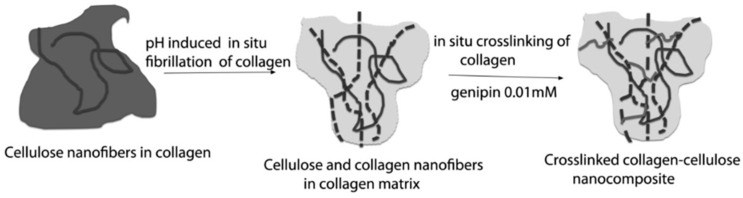
Schematic representation of the preparation of nanocomposites by pH-induced fibrillation with and without crosslinking. The figure above was adapted with permission from the cited reference [83].

**Figure 5 materials-13-01245-f005:**
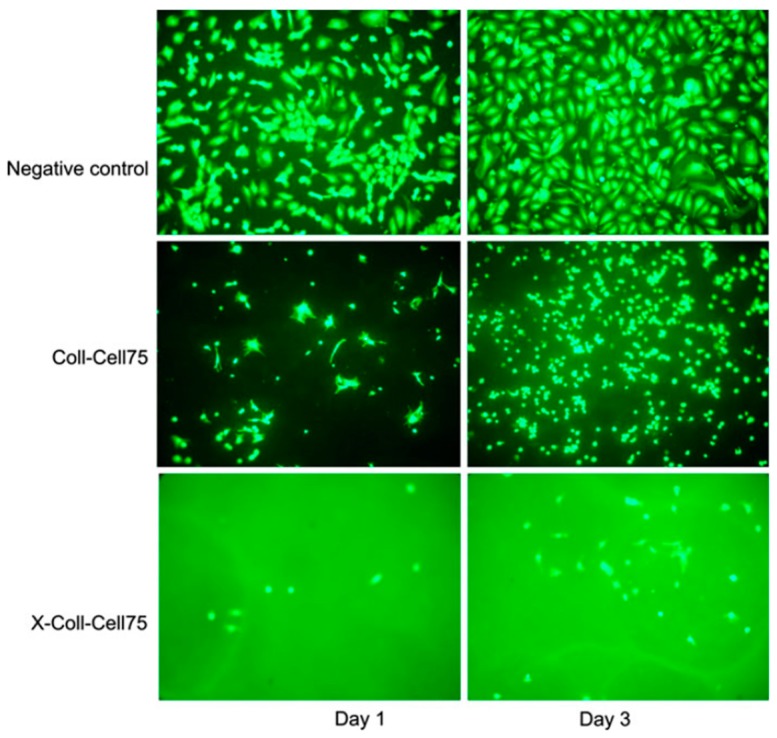
Cell adhesion and growth on the nanocellulose hydrogel compared to the negative control. X: Cross-linker agent, Coll: Collagen, Cell75: 75 wt% of nanocellulose The figure above was adapted with permission from the cited reference [81].

**Figure 6 materials-13-01245-f006:**
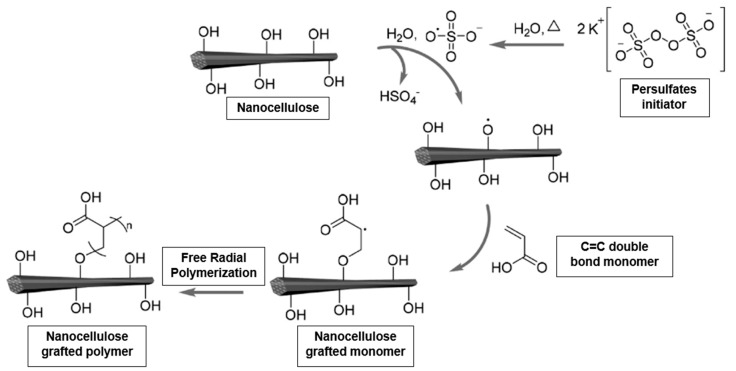
Synthesis of nanocellulose grafted with a monomer via surface-initiated free radical polymerization (modified from [84]).

**Figure 7 materials-13-01245-f007:**
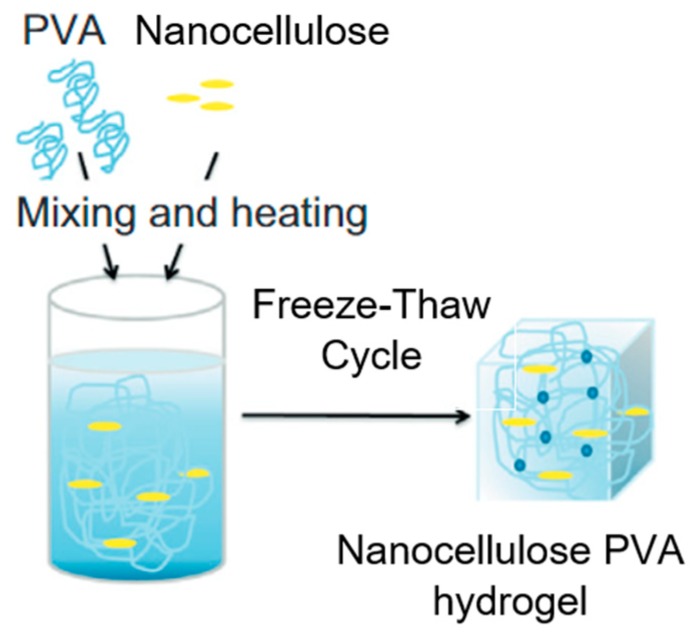
Simple process of nanocellulose hydrogel production (modified from [115]).

**Figure 8 materials-13-01245-f008:**
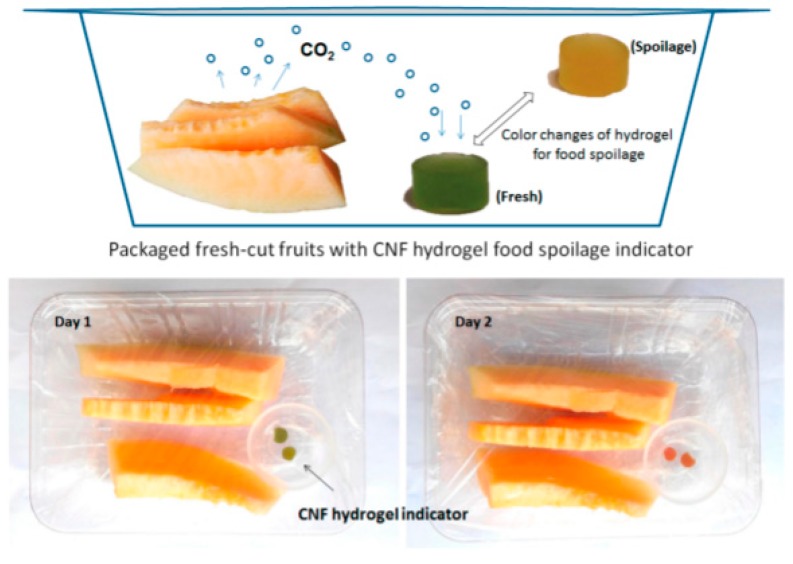
Color changes detected by the cellulose nanofibrils hydrogel indicator. The figure above was adapted with permission from the cited reference [131].

**Table 2 materials-13-01245-t002:** Three type of 3D printing hydrogel processing and its limitations [120].

3D Printing System	Inkjet Printer Based-3D Printing Systems	Nozzle Based-3D Printing Systems	Laser Based-3D Printing Systems
Schematic Representation	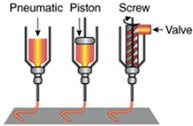	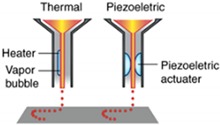	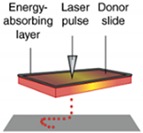
Material Viscosities	30−(6 × 10^7^) mPa/s	3.5–12 mPa/s	1–300 mPa/s
Gelation Method	Chemical Photo-cross-linking Shear Thinning Temperature	Chemical Photo-cross-linking	Chemical Photo-cross-linking
Preparation Time	Low to Medium	Low	Medium to High
Print Speed	Slow (10–50 µm/s)	Fast (1–10,000 droplet/s)	Medium-fast (200–1600 mm/s)
Resolution	5 µm to millimeters wide	<1 pL to >300 pL 50 µm wide	Microscale Resolution
Cell Viability	40–80%	>85%	>95%
Cell Density	High, cell spheroids	Low, <10^6^ cells/mL	Medium, 10^8^ cells/mL
Printer Cost	Medium	Low	High

**Table 3 materials-13-01245-t003:** Hydrogel made from the OPEFB cellulose and its potential applications.

Materials	Potential Applications	References
Microcrystalline cellulose (MCC) extracted from OPEFB, stalks and spikelet	Spikelet MCC—biocomposite Stalk MCC—food and pharmaceutical products	[133]
OPEFB cellulose + NaOH/urea solvent + sodium carboxymethylcellulose (NaCMC)	Tissue engineering and medium for controlled/slow release fertilizer	[134]
OPEFB cellulose + NaOH/urea solvent + NaCMC	Alternative medium for constant water supply for plants	[24]
OPEFB + graphene oxide (GO)	Thermal insulating	[135]

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
