# Peer review of "Potential of Oil Palm Empty Fruit Bunch Resources in Nanocellulose Hydrogel Production for Versatile Applications: A Review"

_materials, 2020, doi:10.3390/ma13051245_

Round 1
Reviewer 1 Report
This paper presents a review of nanocellulose hydrogel from oil palm empty fruit bunch. The manuscript is relatively well written, although there are some weak points that need to be addressed prior to publication. As a consequence, the manuscript needs major corrections.
My main concern about this paper comes from the inconsistency of the topic and nanocellulose hydrogel section (Page 9-21). The authors did a great job introducing OPEFB and methods to produce nanocellulose in the first 9 pages, then the focus of the article transferred to nanocelluloses hydrogels while the connection between nanocellulose hydrogels and OPEFB were hardly mentioned in this section. The author should include more examples and advantages of CNF/CNC obtained from OPEFB applied in nanocellulose hydrogel production, agricultural, and food sensor applications.
Following are the comments I listed to be considered by the authors:
(Page 1, Line 2) The topic does not match with the content introduced in the review. The superabsorbent nature of OPEFB hydrogels was only mentioned a few times without an exclusive introduction.
(Page 2, Line 80) The author should mention the difference in the mechanisms of hydrogel reinforcement. Generally, CNF/CNC can either be incorporated to form a fiber-reinforcement composite system or interpenetrating polymer networks to increase the mechanical strength of the hydrogels.
(Page 5, Line 197) Reference is missing for each of the polymer hydrogels formed by the crosslinking agent. Also, some polymer can form hydrogel via the physical entanglement of polymer chains like PVA via the freeze-thaw method as mentioned in the freeze-thaw method.
(Page 7, Line 269) As far as I see, bacteria nanocellulose is not directly obtained from OPEFB. If that’s true, the author should remind readers about this.
(Page 9, Line 357) “The NCC is applied as reinforcement material in composite and exhibited good results.” Instead of just say good results, the author should list specific data to support his statement. How much NCC was used for reinforcement and how much percentage of increase in mechanical strength was obtained?
(Page 10, Line 403) Any data or literature that reported the comparison of the processes from the point of damage to the hydrogel?
(Page 11, Line 412) For free radical polymerization with CNF mixed with monomers, CNF is regarded as a reinforcement material that is not reactive with monomers although there might be a very small number of radicals transferred to CNF chains. The author should clarify that CNF or CNC does not have reactive vinyl groups to be reactive in FRP and the success in CNF involved FRP is due to radical transfer or controlled radical polymerization (ATRP, RAFT, et al.).
(Page 12, Line 439) “One of the most commonly used approaches for the grafting of polymers from the surface of nanocellulose is surface-initiated free radical polymerization [72]” This is not true. The original statement from Ref 72 is “One of the most commonly used approaches for the grafting of vinyl polymers from the surface of CNCs is surface-initiated free radical polymerization.” It would be great if the author can change the subtitle of 5.2 from “Free Radical Polymerisation Processing Method for Nanocellulose Hydrogel” to “Grafting of nanocellulose hydrogel” and introduce a few more polymer modification of nanocellulose via ring-open polymerization and controlled radical polymerization (ATRP RAFT et al.)
(Page 14, Line 462) Only one article out of 10 is related to nanocellulose and the rest is about PVA in this section. The freezing method enhances the physical cross-linking and entanglement of PVA. However, the author should list more examples and discussion about how it works with nanocellulose hydrogel rather than PVA.
(Page 18, Line 564) If BNC is not obtained from OPEFB then how it is related to the topic? Also “food” may be a confusing subtitle here since CO2 detector is a sensor, not food.
(Page 18, Line 607) Again, bacterial nanocellulose.
(Page 20, Line 641) “biocomposites” seems to be an overlapping concept with biomedical and tissue engineering. Also, it is superabsorbent hydrogel that was introduced rather than biocomposites in this section.
(Page 22, Line 717) The focus of the conclusion has been switched to nanocellulose rather than OPFEB nanocellulose. The author should summarize the advantage, uniqueness, and challenges of nanocellulose obtained from OPFEB discussed in this section.
Some formatting error:
(Page 5, Line 174) References are needed for each of the methods. (Page 7, Line 240) References are missing for Paako, Fall, Abe’s research. (Page 7, Line 258) grammar check needed.
(Page 7, Line 278) Thickness is not an appropriate word to describe nanofibers. Do you mean fiber diameter or width? "nanofibrillated cellulose (NFC)" and "cellulose nanofibres (CNF)" appear numerous times and can be simplified by NFC and CNF.
(Page 10, Line 401) References are needed.
Author Response
(Page 1, Line 2) The topic does not match with the content introduced in the review. The superabsorbent nature of OPEFB hydrogels was only mentioned a few times without an exclusive introduction. We have revised the title of the manuscript in line 2.(Page 2, Line 80) The author should mention the difference in the mechanisms of hydrogel reinforcement. Generally, CNF/CNC can either be incorporated to form a fiber-reinforcement composite system or interpenetrating polymer networks to increase the mechanical strength of the hydrogels. It is a fibre-reinforcement composite where the nanocellulose material will be reinforced into the polymer to form the composite as stated in line 84-86.
(Page 5, Line 197) Reference is missing for each of the polymer hydrogels formed by the crosslinking agent. Also, some polymer can form hydrogel via the physical entanglement of polymer chains like PVA via the freeze-thaw method as mentioned in the freeze-thaw method. We have revised the sentences and added the references in line 172-173.
(Page 7, Line 269) As far as I see, bacteria nanocellulose is not directly obtained from OPEFB. If that’s true, the author should remind readers about this. We have emphasized the information regarding the bacteria nanocellulose is not directly obtained from OPEFB in line 284-285.
(Page 9, Line 357) “The NCC is applied as reinforcement material in composite and exhibited good results.” Instead of just say good results, the author should list specific data to support his statement. How much NCC was used for reinforcement and how much percentage of increase in mechanical strength was obtained? We have quantitatively explained on NCC as reinforcement material which exhibited good results in line 373-375.
(Page 10, Line 403) Any data or literature that reported the comparison of the processes from the point of damage to the hydrogel? There is no study work on comparison of the processes from the point of damage to the hydrogel. But there is study on the effect of preparation temperature on hydrogel. This have been included in the manuscript in line 424-428.
(Page 11, Line 412) For free radical polymerization with CNF mixed with monomers, CNF is regarded as a reinforcement material that is not reactive with monomers although there might be a very small number of radicals transferred to CNF chains. The author should clarify that CNF or CNC does not have reactive vinyl groups to be reactive in FRP and the success in CNF involved FRP is due to radical transfer or controlled radical polymerization (ATRP, RAFT, et al.). We have clarified that CNF or CNC does not have reactive vinyl groups to be reactive in FRP and the success in CNF involved FRP is due to radical transfer or controlled radical polymerization in line 437-438.
(Page 12, Line 439) “One of the most commonly used approaches for the grafting of polymers from the surface of nanocellulose is surface-initiated free radical polymerization [72]” This is not true. The original statement from Ref 72 is “One of the most commonly used approaches for the grafting of vinyl polymers from the surface of CNCs is surface-initiated free radical polymerization.” It would be great if the author can change the subtitle of 5.2 from “Free Radical Polymerisation Processing Method for Nanocellulose Hydrogel” to “Grafting of nanocellulose hydrogel” and introduce a few more polymer modification of nanocellulose via ring-open polymerization and controlled radical polymerization (ATRP RAFT et al.) We have changed the subtitle of 5.2 as suggested to Grafting of nanocellulose hydrogel and more polymer modification have been discussed in line 436.
(Page 14, Line 462) Only one article out of 10 is related to nanocellulose and the rest is about PVA in this section. The freezing method enhances the physical cross-linking and entanglement of PVA. However, the author should list more examples and discussion about how it works with nanocellulose hydrogel rather than PVA. More discussion regarding the effect of nanocellulose toward the freezing and thawing process have been included. Part of the process highlighting PVA were removed.
(Page 18, Line 564) If BNC is not obtained from OPEFB then how it is related to the topic? Also “food” may be a confusing subtitle here since CO2 detector is a sensor, not food. We have revised and restructured all the paragraph in subtitle 6.0 to be more relatable with the manuscript title. Therefore, we have removed the unrelated information that might be misleading.
(Page 18, Line 607) Again, bacterial nanocellulose. We have revised and restructured all the paragraph in subtitle 6.0 to be more relatable with the manuscript title. Therefore, we have removed the unrelated information that might be misleading.
(Page 20, Line 641) “biocomposites” seems to be an overlapping concept with biomedical and tissue engineering. Also, it is superabsorbent hydrogel that was introduced rather than biocomposites in this section. We have revised and restructured all the paragraph in subtitle 6.0 to be more relatable with the manuscript title. Therefore, we have removed the unrelated information that might be misleading.
(Page 22, Line 717) The focus of the conclusion has been switched to nanocellulose rather than OPFEB nanocellulose. The author should summarize the advantage, uniqueness, and challenges of nanocellulose obtained from OPFEB discussed in this section. We have revised the conclusion to fit the main scope.
(Page 5, Line 174) References are needed for each of the methods. (Page 7, Line 240) References are missing for Paako, Fall, Abe’s research. (Page 7, Line 258) grammar check needed. We have added the related references for line 245, 249 and 252, respectively.
(Page 7, Line 278) Thickness is not an appropriate word to describe nanofibers. Do you mean fiber diameter or width? "nanofibrillated cellulose (NFC)" and "cellulose nanofibres (CNF)" appear numerous times and can be simplified by NFC and CNF. We have revised the word thickness with width in line 292 and have standardized the terminology for cellulose nanofiber as CNF throughout the manuscript.
(Page 10, Line 401) References are needed. The sentence has been removed in revise version to fit the topic of OPEFB nanocellulose hydrogel.
Reviewer 2 Report
On the positive side: this is an informative and well-written review paper on an important subject.
On the less-positive side: unfortunately, the authors did not narrow the scope of their paper to the area specified in the title of their work (the possible use of palm empty fruit bunch as a raw material for the production of superabsorbent nanocellulose hydrogel), and produced a very wide overview on the general topic of nanocellulose hydrogels. Since there are a number of excellent reviews and books available in this field, the authors do not provide a clear reason why a new one needs to be published.
My suggestions related to the paper are:
1) A thorough linguistic review has to be done. To illustrate the importance of this task, I marked a number of problematic phrases and expressions on the first few pages of the manuscript.
2) As indicated previously, the paper has to focus on the use of palm empty fruit bunch. Therefore, the paper needs to be rewritten in a significantly shorter and targeted form: the lengthy introduction (and with it most of the figures) can be omitted from the paper.
3) The goals of the research need to be specified clearly.
4) Only those literature findings need to be cited that are consistent with the goals.
5) Novel ideas need to be presented, and contradictions in the literature, as well as missing points, need to be identified and solutions need to be suggested.
6) The figures seem to be taken from other publications: permissions for reproduction need to be given, when necessary.
PS: I do not have access to plagiarism checker software.

Author Response
A thorough linguistic review has to be done. To illustrate the importance of this task, I marked a number of problematic phrases and expressions on the first few pages of the manuscript. We have revised all the related phrases and expressions mentioned.As indicated previously, the paper has to focus on the use of palm empty fruit bunch. Therefore, the paper needs to be rewritten in a significantly shorter and targeted form: the lengthy introduction (and with it most of the figures) can be omitted from the paper. We tried our best to shorten the introduction part.
The goals of the research need to be specified clearly. We have emphasized our goal in line 87-88.
Only those literature findings need to be cited that are consistent with the goals. Our literatures are consistent with our goal.
Novel ideas need to be presented, and contradictions in the literature, as well as missing points, need to be identified and solutions need to be suggested. We have taken into consideration of all the suggestions by the reviewers and revised our manuscript accordingly.
The figures seem to be taken from other publications: permissions for reproduction need to be given, when necessary.
PS: I do not have access to plagiarism checker software.
We have obtained the permissions from the cited authors and added in the caption for every figure.Reviewer 3 Report
The authors reviewed use of different forms of nanomaterials obtained from empty fruit bunch for various applications. The manuscript presents representative examples of nanocellulose materials obtained from EFB or OPEFB. The authors particularly highlight the regenerated cellulose, nanocellulose for various applications. In general, the topic of the manuscript is appropriate to the review for Materials. However, there is a critical concern. The main direction/structure of the current version of the manuscript should be revised. I don’t see many cases reviewed in this manuscript, regarding “nanocellulose hydrogel” from OPEFB. I might be wrong, but I found one example (line 303). Otherwise, most of the reviewed articles are about nanocellulose hydrogels, which are not necessarily obtained from “OPEFB.” This major problem of the manuscript should be corrected for being published in Materials.
Author Response
The main direction/structure of the current version of the manuscript should be revised. I don’t see many cases reviewed in this manuscript, regarding “nanocellulose hydrogel” from OPEFB. I might be wrong, but I found one example (line 303). There is only one paper regarding nanocellulose hydrogel from OPEFB which is cited as reference [69]. Therefore, we have revised our title to discuss the potential of OPEB as a raw material in nanocellulose hydrogel.
Reviewer 4 Report
in general: The work presented is a decent review of the field. It is well structured and presented. In this reviewer's opinion it should be published, although there are various points which should and/or must be addressed before publication however. I will go through these below.
Figures:
If the figure is copied from a reference or based on data from reference this must be explicitly stated in the caption. Whether licensing permits its use or if special permission has been obtained should of course be mentioned explicitly as well.
The authors might want to check the resolution (DPI) of the images used in figures. Some seem a little pixelated - and compression artefacts are evident in some, e.g. Figure 14.
Figure 1 is visually appealing, but does not appear to add much quantitative information. This reviewer would consider accepting the figure as-is, but also thinks the figure could contribute substantially more to the paper if it visualized relative production volumes. Ideally as a bar chart or similar visualization.
Figure 2: Consider using a stacked bar chart instead of line and bar chart. This reviewer would also consider changing the unit on the Y-axis to avoid so many zeroes. e.g. by using mn hectare as opposed to hectare. I would consider the annotations excessive as the specific values are explicitly mentioned in the cited reference.
Figure 6: No scale bar is supplied.
Figure 12: I am not sure how much this contributes to the text.
Figure 14: I would consider splitting this into two stacked graphs and not have all data presented in the same axial environment - the presented data
Scientific:
A more thorough discussion of production techniques could be beneficial. The review deals more with applications, so I am not suggesting a new sub-chapter, or even a large paragraph, but as CNF is the core of the work it would perhaps be beneficial to discuss morphology and pre-treatments briefly with references to other, relevant reviews (plural).
Line 60: The n should be subscript.
Line 300-301 Crystallinity index, this is presumably measured by WAXS, but it is not specified. The result obtained for CrI depends heavily on the method used, please specify.
Line 309 - what units are used? Hours? Days? Minutes? Please specify
Line 316-317 what is meant by this?
Line 536-537 Who reported this, and where is it published? Please specify.
Line 579-582 - This claim of health benefit should be clarified and must be well substantiated, ideally with multiple sources.
While not a major point in the work, SEM micrographs of hydrogels are used, but the method is not discussed. There are some contentious points to most SEM work on hydrogels - see e.g. Torstensen et al. 2018 DOI: 10.1007/s10570-018-1854-8
Lines 59-60 - it is the crystal allomorph that is known as cellulose II. Chemically it is identical to cellulose I (which occurs naturally and exists as both alpha and beta allomorphs) but the polymer strands are anti-parallel in cellulose II and parallel in cellulose I. A source I recommend citing on this is Zugenmaier's book (2008, ISBN 9783540739333). Stiffness of the crystalline areas may be higher, but not infrequently regenerated cellulose is also rather amorphous. Stating that it is stronger should be substantiated. This should be discussed a little more.
The chemistry of cellulose is worth expounding upon briefly. While the chemical formula has merit, it tells nearly nothing of the chemistry involved, the structure of cellulose is paramount and should be mentioned. The polymer is linear and planar, consisting of ß 1-4 linked glucose monomers (not cellobiose as is frequently stated). A. French 2017 (DOI: 10.1007/s10570-017-1450-3) is worth citing as there has been some discussion on this subject.
Language:
The language in the paper is inadequate at the time of writing and needs a major rework. There are several mistakes, linguistically speaking. Besides minor typos a more thorough rework of the manuscript is necessary as the intended message is sometimes difficult to grasp. Mistakes vary in severity and frequency across the manuscript. A brief list containing examples of mistakes follow (note, this is far from exhaustive):
Line 56: wide, not wide's.
Line 259:'are also can be'?
Line 421: 'have wrote' is not correct. 'wrote' is.
Line 495 weightage?
Line 644 I did not understand.
Several times the authors say something is "nanocellulose contented" - which does not mean nanocellulose is contained within something - rather that the something is satisfied/pleased by nanocellulose.
Acronyms:
Further, sometimes acronyms are specified several times, are not defined or used interchangably. e.g.:
CNF is defined on lines 76, 268, 275 and in Table 3.
OPEFD is defined on line 14, 51, 108, 133, 140.
CNC is also defined multiple times. In contrast TCF is not defined and EPF, while evident from context, is not defined explicitly either.
CNF and NFC are both used for CNF. CNF is preferred nomenclature according to ISO/TS 20477:2017 and TAPPI’s proposed WI 3021.
MCC is defined - but what the difference between CNC and MCC is is not specified. As this is a review the audience might be wide, and the distinction should be clarified.
Please correct and double check.
All in all a good piece of work. I wish the authors best of luck with revisions and future work.
Author Response
If the figure is copied from a reference or based on data from reference this must be explicitly stated in the caption. Whether licensing permits its use or if special permission has been obtained should of course be mentioned explicitly as well. We have obtained the permissions from the cited authors and added in the caption for every figure.
The authors might want to check the resolution (DPI) of the images used in figures. Some seem a little pixelated - and compression artefacts are evident in some, e.g. Figure 14. The figure has been removed in revise version to fit the topic of OPEFB nanocellulose hydrogel. We have made sure that the images used are in high resolution.
Figure 1 is visually appealing, but does not appear to add much quantitative information. This reviewer would consider accepting the figure as-is, but also thinks the figure could contribute substantially more to the paper if it visualized relative production volumes. Ideally as a bar chart or similar visualization. We have removed the Figure 1 as the similar information already mentioned in the text.
Figure 2: Consider using a stacked bar chart instead of line and bar chart. This reviewer would also consider changing the unit on the Y-axis to avoid so many zeroes. e.g. by using mn hectare as opposed to hectare. I would consider the annotations excessive as the specific values are explicitly mentioned in the cited reference. We have revised the bar chart into stacked bar chart for new Figure 1.
Figure 6: No scale bar is supplied. The original Figure that we are cited does not has the scale bar in new Figure 5.
Figure 12: I am not sure how much this contributes to the text. We have removed the Figure 12.
Figure 14: I would consider splitting this into two stacked graphs and not have all data presented in the same axial environment - the presented data The figure has been removed in revise version to fit the topic of OPEFB nanocellulose hydrogel.
Line 60: The n should be subscript. We have changed the n into subscript format in line 63.
Line 300-301 Crystallinity index, this is presumably measured by WAXS, but it is not specified. The result obtained for CrI depends heavily on the method used, please specify. We added the equipment that measure CrI in line 314-315.
Line 309 - what units are used? Hours? Days? Minutes? Please specify We have added the unit in line 324.
Line 316-317 what is meant by this? We have elaborated the statement in line 330-333.
Line 536-537 Who reported this, and where is it published? Please specify. The paragraph has been removed in revise version to fit the topic of OPEFB nanocellulose hydrogel.
Line 579-582 - This claim of health benefit should be clarified and must be well substantiated, ideally with multiple sources. The paragraph has been removed in revise version to fit the topic of OPEFB nanocellulose hydrogel.
While not a major point in the work, SEM micrographs of hydrogels are used, but the method is not discussed. There are some contentious points to most SEM work on hydrogels - see e.g. Torstensen et al. 2018 DOI: 10.1007/s10570-018-1854-8 We have added the related information in line 405-407.
Lines 59-60 - it is the crystal allomorph that is known as cellulose II. Chemically it is identical to cellulose I (which occurs naturally and exists as both alpha and beta allomorphs) but the polymer strands are anti-parallel in cellulose II and parallel in cellulose I. A source I recommend citing on this is Zugenmaier's book (2008, ISBN 9783540739333). Stiffness of the crystalline areas may be higher, but not infrequently regenerated cellulose is also rather amorphous. Stating that it is stronger should be substantiated. This should be discussed a little more. We have removed the word stronger and revised the sentences with additional information on cellulose I and II as you suggested in line 65-66.
The chemistry of cellulose is worth expounding upon briefly. While the chemical formula has merit, it tells nearly nothing of the chemistry involved, the structure of cellulose is paramount and should be mentioned. The polymer is linear and planar, consisting of ß 1-4 linked glucose monomers (not cellobiose as is frequently stated). A. French 2017 (DOI: 10.1007/s10570-017-1450-3) is worth citing as there has been some discussion on this subject. We have added the related information as you suggested in line 59-62.
Line 56: wide, not wide's. We have corrected the word wide in line 55
Line 259:'are also can be'? We have removed the word.
Line 421: 'have wrote' is not correct. 'wrote' is. We have corrected the related word in line 450
Line 495 weightage? We have rephrased the sentence in line 509
Line 644 I did not understand.
Several times the authors say something is "nanocellulose contented" - which does not mean nanocellulose is contained within something - rather that the something is satisfied/pleased by nanocellulose. The words have been rephrased in manuscript in line 403 and 504.
Further, sometimes acronyms are specified several times, are not defined or used interchangably. e.g.:
CNF is defined on lines 76, 268, 275 and in Table 3.
OPEFD is defined on line 14, 51, 108, 133, 140.
CNC is also defined multiple times. In contrast TCF is not defined and EPF, while evident from context, is not defined explicitly either.
CNF and NFC are both used for CNF. CNF is preferred nomenclature according to ISO/TS 20477:2017 and TAPPI’s proposed WI 3021.
MCC is defined - but what the difference between CNC and MCC is is not specified. As this is a review the audience might be wide, and the distinction should be clarified.
Please correct and double check.
Round 2
Reviewer 1 Report
Since all my comments have been properly addressed by the authors, I would like to suggest acceptance of this article in the present form.
Author Response
Since all my comments have been properly addressed by the authors, I would like to suggest acceptance of this article in the present form
Response: We appreciate your previous useful comments.
Reviewer 2 Report
The paper improved significantly. Unfortunately, it is not yet publishable in its current form.
Although the authors claim that a thorough linguistic review has to be done, the English used in the revision is still quite poor making the paper difficult to read and comprehend (I marked a number of linguistic problems on the first three pages - see attached file).
In addition, the authors failed to justify the publication of the paper: in its current form, it is still hardly more than a compilation of the available literature from the field - without critical evaluation and original ideas.

Author Response
The paper improved significantly. Unfortunately, it is not yet publishable in its current form.
Although the authors claim that a thorough linguistic review has to be done, the English used in the revision is still quite poor making the paper difficult to read and comprehend (I marked a number of linguistic problems on the first three pages - see attached file).
In addition, the authors failed to justify the publication of the paper: in its current form, it is still hardly more than a compilation of the available literature from the field - without critical evaluation and original ideas.
Response: We have sent our manuscript to proof-reader for further English check. Thank you very much for your kind suggestion. We also added several critical evaluation sentences in line 152-154, 255-259, 287-296, and 540-551.
Reviewer 3 Report
The authors claim that the title of the manuscript has been revised since the potential of OPEB is discussed as a raw material in nanocellulose in the manuscript. However, the type of the manuscript is "a review." According to the definition of reviews as types of publications on the MDPI website, “reviews provide concise and precise updates on the latest progress made in a given area of research.” Therefore, the authors should have reviewed a reasonable number of articles that directly/closely related to “nanocellulose hydrogel from OPEFB”. To avoid any confusion and discrepancy of the manuscript contribution, my suggestion is to remove “hydrogel” term from the current title and to include the nanocellulose hydrogel paper in Challenges and Future Direction section as potential of OPEB as a raw material in hydrogel. This is reasonable to follow the manuscript without major revision.
Author Response
The authors claim that the title of the manuscript has been revised since the potential of OPEB is discussed as a raw material in nanocellulose in the manuscript. However, the type of the manuscript is “a review.” According to the definition of reviews as types of publications on the MDPI website, “reviews provide concise and precise updates on the latest progress made in a given area of research.” Therefore, the authors should have reviewed a reasonable number of articles that directly/closely related to “nanocellulose hydrogel from OPEFB”. To avoid any confusion and discrepancy of the manuscript contribution, my suggestion is to remove “hydrogel” term from the current title and to include the nanocellulose hydrogel paper in Challenges and Future Direction section as potential of OPEB as a raw material in hydrogel. This is reasonable to follow the manuscript without major revision.
Response: Thank you very much for your suggestion. As the title implies, we discussed on the “potential” of OPEFB nanocellulose as feedstock for hydrogel. We aware that the literatures regarding OPEFB hydrogel is limited and therefore our starting point is from the identification of OPEFB as potential feedstock. In the paper, we discussed the availability of OPEFB in the country, preparation method of hydrogel and the potential applications of OPEFB as well as challenges and future direction. Therefore, we are aiming that innovative ideas can be generated by the readers while reading this paper in order to begin new research in this area. We think this is the purpose of doing a review paper. In our opinion, the hydrogel is the vital part of this paper that define our novelty. This is because the demand of hydrogel getting higher (line 107) for high-end applications. Thus, abundant availability of OPEFB has great potential to cater the high demand for nanocellulose hydrogel.
Reviewer 4 Report
I thank the authors for their work.
The manuscript has been improved relative the first submission, but there are still issues which need rectifying. These are - as I see it - primarily of a linguistic nature. There are issues with the English which are of such a character that the authors should perform a thorough reworking. should the authors not feel their linguistic mastery is up to par, external help should be acquired.
The results can be unfortunate. I listed a few examples in the previous run, and these are thankfully corrected, but a general reworking is, unfortunately, necessary. An example or two will follow. On line 63 the authors state that the chemical formula for cellulose is similar between cellulose I and II. This is not the case, it is identical. The formula is identical irrespective of allotrope -cellulose I, II, III, IV and subtypes. They are not similar, but identical. The differences between the types is not in composition but structure. This is not, in the current form, clear. From the authors formulation it appears there are stoichiometric differences between the allotropes. There is not. The molecular formula is identical. The use of similar conflicts with previous text, so I am sure it is an oversight.
In the paragraph beginning on line 101 the authors start with a long and somewhat poor sentence. I expect the authors mean 'greenhouse gas' when they say 'green gas', and the 'the' in `the human health' is superfluous. I would also presume that the end of the sentence: "an abundant available nature-friendly materials as alternative for fossil-fuel based products" refers to either one specific or several environmentally friendly material or materials (the authors mix singular and plural) which is either abundant or abundantly available. I would also assume that the authors refer not to material based on fuels, which is what they say, but rather petrochemicals - or fossil materials - given that the word fuel implies it is burned or otherwise consumed for energy production.
The above examples are, as far as I can tell, fairly representative for the text as a whole. The manuscript, as a whole, must therefore be thoroughly reworked prior to publication. Scientifically, however, the paper is - in my opinion - ready for publication (though the chemical issue described above does need to be corrected).
Best of luck to the authors in the reworking of the manuscript.
Author Response
I thank the authors for their work.
The manuscript has been improved relative the first submission, but there are still issues which need rectifying. These are - as I see it - primarily of a linguistic nature. There are issues with the English which are of such a character that the authors should perform a thorough reworking. should the authors not feel their linguistic mastery is up to par, external help should be acquired.
The results can be unfortunate. I listed a few examples in the previous run, and these are thankfully corrected, but a general reworking is, unfortunately, necessary. An example or two will follow. On line 63 the authors state that the chemical formula for cellulose is similar between cellulose I and II. This is not the case, it is identical. The formula is identical irrespective of allotrope -cellulose I, II, III, IV and subtypes. They are not similar, but identical. The differences between the types is not in composition but structure. This is not, in the current form, clear. From the authors formulation it appears there are stoichiometric differences between the allotropes. There is not. The molecular formula is identical. The use of similar conflicts with previous text, so I am sure it is an oversight.
In the paragraph beginning on line 101 the authors start with a long and somewhat poor sentence. I expect the authors mean 'greenhouse gas' when they say 'green gas', and the 'the' in `the human health' is superfluous. I would also presume that the end of the sentence: "an abundant available nature-friendly materials as alternative for fossil-fuel based products" refers to either one specific or several environmentally friendly material or materials (the authors mix singular and plural) which is either abundant or abundantly available. I would also assume that the authors refer not to material based on fuels, which is what they say, but rather petrochemicals - or fossil materials - given that the word fuel implies it is burned or otherwise consumed for energy production.
The above examples are, as far as I can tell, fairly representative for the text as a whole. The manuscript, as a whole, must therefore be thoroughly reworked prior to publication. Scientifically, however, the paper is - in my opinion - ready for publication (though the chemical issue described above does need to be corrected). Best of luck to the authors in the reworking of the manuscript
Response: We have sent our manuscript to proof-reader for further English check. Thank you very much for your explanation and suggestions. We also addressed the issue in line 64-66 regarding cellulose I and II where it is indeed identical in terms of chemical formulation and only differ in structure arrangement where cellulose I has parallel arrangement structure whereas cellulose II has antiparallel structure.